# Deciphering the Signaling Mechanisms of Osteosarcoma Tumorigenesis

**DOI:** 10.3390/ijms241411367

**Published:** 2023-07-12

**Authors:** Bikesh K. Nirala, Taku Yamamichi, Jason T. Yustein

**Affiliations:** Aflac Cancer and Blood Disorders Center, Emory University, Atlanta, GA 30322, USA; bikesh.kumar.nirala@emory.edu (B.K.N.); taku.yamamichi@emory.edu (T.Y.)

**Keywords:** osteosarcoma, signaling pathways, oncogenes, tumor suppressors, immunotherapy

## Abstract

Osteosarcoma (OS) is the predominant primary bone tumor in the pediatric and adolescent populations. It has high metastatic potential, with the lungs being the most common site of metastasis. In contrast to many other sarcomas, OS lacks conserved translocations or genetic mutations; instead, it has heterogeneous abnormalities, including somatic DNA copy number alteration, ploidy, chromosomal amplification, and chromosomal loss and gain. Unfortunately, clinical outcomes have not significantly improved in over 30 years. Currently, no effective molecularly targeted therapies are available for this disease. Several genomic studies showed inactivation in the tumor suppressor genes, including *p53*, *RB*, and *ATRX,* and hyperactivation of the tumor promoter genes, including *MYC* and *MDM2*, in OS. Alterations in the major signaling pathways, including the PI3K/AKT/mTOR, JAK/STAT, Wnt/β-catenin, NOTCH, Hedgehog/Gli, TGF-β, RTKs, RANK/RANKL, and NF-κB signaling pathways, have been identified in OS development and metastasis. Although OS treatment is currently based on surgical excision and systematic multiagent therapies, several potential targeted therapies are in development. This review focuses on the major signaling pathways of OS, and we propose a biological rationale to consider novel and targeted therapies in the future.

## 1. Introduction

Osteosarcoma (OS) is a rare cancer arising from the mesenchymal cells forming the bone. It is the most common and highly metastatic bone tumor in children and adolescents [1]. Its incidence is higher in adolescents (0.8–1.1/100,000/year in the age group of 15–19 years), with a second peak in older adults [2,3]. Nearly two-thirds of primary tumors occur near the knee joint, with the most common sites being the distal femur, proximal humerus and proximal tibia [4]. Despite the extensive genomic aberrations, OS has no pathognomonic DNA translocation or targetable mutations [5]. Thus, no effective molecularly targeted therapies for OS are currently available. The diagnosis of OS is based on morphological characteristics since no specific molecular markers or testing are available in clinical practice. The management of OS is challenging and requires a multidisciplinary approach. Surgical excision and systematic multiagent therapy are standard clinical practices for OS treatment. However, there is a pressing need to identify novel therapeutic approaches and biomarkers to manage the disease better, given the high relapse rate and poor prognosis of metastatic disease. One of the critical factors in OS development is chromosomal instability and genetic changes [6]. Oncogenes and tumor suppressor genes are often affected in OS. The immune system also plays a role in regulating tumor growth and propagation, and it is evident that the tumor-infiltrating immune cells contribute to the metastatic cascade [7]. Tumor metastasis is the primary challenge for OS therapy [8]. The five-year survival rate of OS has increased to about 70% since the 1970s, although it is only 20–30% for patients with metastasis [9]. Most OSs infiltrate the surrounding tissue and metastasize to the lung.

A better understanding of the bone microenvironment, the interaction between the tumor and non-tumor cells, and the mechanism of OS metastasis will help find a therapeutic target for OS. Several major signaling pathways have been identified in OS tumor development and metastasis, including the PIK3, JAK/STAT, Wnt/β-catenin, NOTCH, Hedgehog, Ras, TGF-β, MAPK/AKT/mTOR, RANK/RANKL, and NF-κB signaling pathways [10,11]. This review emphasizes the current understanding of the signaling pathways involved in OS tumor development and metastasis. Moreover, we also highlight the role of chromosomal instability and immune regulation in OS tumorigenesis and metastasis.

## 2. Molecular Abnormalities in Osteosarcoma

### 2.1. Chromosomal Abnormalities

OS is a genetically complex and heterogeneous tumor characterized by chromosomal instability and genetic alterations that lead to aneuploidy and increased tumor aggressiveness. The high rates of chromosomal rearrangements in OS include structural chromosomal abnormalities, such as translocations, deletions, amplifications, and chromothripsis, an extreme form of chromosomal instability [12]. Chromothripsis is associated with increased genomic instability and tumor progression, and it frequently occurs in highly aggressive tumors, including OS. Although the exact cause of chromothripsis and its role in tumorigenesis remain unclear, recent genomic studies have revealed that it is context-dependent and occurs at an overall incidence of between 2% and 3% in pan-cancer samples but over 77% in OS and 100% in liposarcoma [13]. Some potential mechanisms underlying chromothripsis are emerging, including the generation of DNA breaks and rejoining of the DNA fragments, generation of micronuclei, premature chromosome condensation, breakage–fusion–bridge cycle and telomere dysfunction, and ionizing radiation [14]. While the exact mechanistic cause of chromothripsis is still undefined, Crasta et al. identified the micronuclei, having many features of primary nuclei, formed from the acentric fragments of chromosomes, which produce DNA damage behind the chromothripsis [15]. Zhang et al., by using a combination of live cell imaging and single-cell genome sequencing, demonstrated that micronucleus formation could indeed generate a spectrum of genomic rearrangements, which recapitulate the features of chromothripsis [16]. Gong et al. described how Ran GTPase-activating protein 1 (RanGAP1) is commonly reduced or inactivated in human OS, leading to a high probability of chromothripsis, which drives tumorigenesis through its direct effects on the spindle-assembly checkpoint and decatenation and secondary effects on DNA damage surveillance [17].

### 2.2. Inactivation of Tumor Suppressor Genes and Amplification of Oncogenes

Inactivation of tumor suppressor genes such as *TP53*, *RB1*, *ATRX*, and *DLG2* is frequently observed in OS, which is thought to be involved in OS tumorigenesis [18]. The *TP53* gene is OS’s most frequently dysregulated gene [18]. Whole-genome DNA sequencing from OS tumor samples demonstrates multiple somatic chromosomal lesions, including structural variations (SVs) and copy number alterations (CNAs). Kataegis is a single nucleotide variation (SNV) detected in 50% of OS tumors. Chen et al. identified p53 pathway lesions in all OS patients, while it was translocated in around 50% of the patients to the first intron of the *TP53* gene, leading to gene inactivation. This mechanism of *TP53* gene inactivation is unique to OS among pediatric cancers [18]. The p53 protein is a tumor suppressor protein involved in DNA damage recognition that induces apoptosis, cellular quiescence, or senescence. Another tumor suppressor gene frequently inactivated in OS is *RB1*, located at chromosome *13q14.2* [19]. *RB1* encodes the tumor suppressor protein pRB, which is vital for preventing cell cycle progression. *ATRX* is an important tumor suppressor in OS, and it is a part of a multiprotein complex that regulates chromatin remodeling, nucleosome assembly, and telomere maintenance. Furthermore, a recent report noted that loss of ATRX promotes OS tumor through increased NF-κB signaling and integrin binding [20]. *DLG2* is a tumor suppressor gene, and its copy number loss occurs in 42% of human and 56% of canine OS [21]. Deletion of *Dlg2* in a mouse model led to the acceleration of OS development [21].

Hyperactivation of tumor-promoting genes such as *MYC* and *MDM2* is associated with OS tumorigenesis. The gain of the *8q24* chromosomal locus, which harbors the oncogene *MYC*, has been reported in several OS patients [22,23]. MYC is involved in cell cycle regulation, protein biogenesis, metabolism, signal transduction, transcription, and translation [24,25]. A recently generated *Myc* knock-in genetically engineered mouse model of an OS tumor not only identified intrinsic Myc-mediated mechanisms of OS tumorigenesis but also identified a novel molecular mechanism through which Myc regulates the profile and function of the OS immune landscape [26,27,28]. The oncoprotein MDM2 is a p53 inhibitor, which promotes p53 degradation and downregulates its transcription. Amplification of *MDM2* (chromosome *12q15*) is more frequent in OS metastasis and recurrence.

### 2.3. Epigenetic Modification in OS Progression

Epigenetic modifications, including DNA methylation, histone acetylation, and methylation, are critical in the pathogenesis of several cancers, including OS [29]. The level of histone H3 lysine trimethylation was reported to be lower in human OS tissue and cell lines compared with normal bone tissue and osteoblast cells. Enhancement of H3 methylation after treatment with the histone lysine demethylase inhibitor 5-carboxy-8-hydroxyquinoline (IOX-1) showed inhibition of OS migratory and invasive capabilities. Enhanced histone H3 lysine trimethylation levels sensitized cisplatin against the cisplatin-resistant (MG63-CR) cells [30]. Previous studies showed enhancement of the expression of the lysine-specific demethylases KDM1A, KDM2B, KDM4A, KDM6A, KDM6B in OS progression [31,32,33,34]. Recently, Twenhafel et al. provided a comprehensive review of recent advances in the epigenetics of OS and highlighted the clinical benefits in the field of OS research [35]. Morrow et al. highlighted the genetic and epigenetic defects in OS and emphasized the role of epigenetic dysregulation in tumor suppression and oncogene regulation [6]. 

## 3. Major Signaling Pathways in OS Tumor Development and Metastasis

Signaling pathways and molecular networks are highly regulated under normal growth conditions and control critical pro-survival and pro-growth cellular processes. Dysregulation of the signaling pathways contributes to OS development, metastasis, and chemoresistance. This review will focus on the essential signaling pathways of OS that play a vital role in disease development and progression. Figure 1 shows the different signaling pathways involved in OS tumorigenesis and metastasis. 

### 3.1. PIK3/AKT/mTOR Pathway

The family of lipid kinases known as phosphoinositide 3-kinases (PI3Ks) is vital to almost all cell and tissue biology aspects and central to human cancer, including glioblastoma, sarcoma, skin, respiratory, digestive and breast cancer [36,37]. PI3Ks are multifunctional and regulate a wide range of signaling, membrane trafficking, cell survival, metabolism, and metastasis [38]. An increasing body of evidence has shown that the PI3K pathway is frequently hyperactivated in OS and contributes to tumorigenesis, proliferation, metastasis, invasion, cell cycle progression, inhibition of apoptosis, angiogenesis, and chemoresistance [39]. The PI3K/AKT pathway can be abnormally triggered by interacting with various growth factor receptors, such as vascular endothelial growth factor receptor (VEGFR), epithelial growth factor receptor (EGFR), and fibroblast growth factor receptor (FGFR), and by mutations in the *PIK3CA*, *AKT*, *PTEN*, and *mTOR* genes [40]. Immunostaining analysis of primary OS shows that activation of the PI3K/AKT signaling pathway facilitates the progression of OS and is also associated with the patient’s poor prognosis. Additionally, the activation of AKT is associated with lung metastasis [41,42,43]. 

AKT phosphorylates several substrates and downstream effectors, including mTOR, matrix metalloproteinase (MMP), cyclin-dependent kinases (CDKs), and VEGF, associated with tumor progression and metastasis [44,45]. The secretion of VEGF-A and FGF2 from OS cells promotes migration and invasion by activating the PI3K and AKT pathways, eventually leading to MMP9 overexpression [46]. Human epidermal growth factor receptor 4 (HER4), a member of the ERBB family, is upregulated in OS tumor tissue and cell lines, promoting OS progression by inactivating the PTEN-PI3K/AKT pathway [47]. VCAM-1 is increased in metastatic OS cells through the activation of the CXCL1-CXCR2/focal adhesion kinase FAK-PI3K-AKT-NF-κB pathway. CXCL1/CXCR2 pathway activation has been a crucial indicator of lung metastasis in OS, and the higher expression of CXCL1 is positively correlated with the migratory and invasive behavior of OS cells [48]. Other examples highlighting the importance of PI3K-AKT mediated signaling in advanced OS include the increased IHC staining of CDC42 effector protein 3 (CDC42EP3) [49], which was associated with patients’ pathological stage and grade, while the zinc finger CCHC domain containing 12 genes (*ZCCHC12*) is highly upregulated at the transcriptomic level in OS compared to normal bone tissues and has a vital role in OS cell proliferation and migration [50]. 

Several non-coding RNAs, including LncRNA H19, LINC00968, LINC00628, LncRNA NDRG1, LncDANCR, and circ_001422, are markedly associated with the OS advanced clinical stage, larger tumor size, higher incidence of metastases, and poorer prognosis, and they have been found to be mediated through the PI3K/AKT signaling pathway [43,51,52,53,54,55].

The PI3K/AKT signaling pathway also contributes to drug resistance in different types of cancers, including lung and esophageal cancer [56]. Several recent reports have shown the involvement of PI3K/AKT pathways in chemoresistance in OS. The zinc transporters Zrt- and Irt-related protein (ZIP/SLC39) ZIP10 is highly expressed in OS, which promotes cell proliferation and chemoresistance that is mediated through the activation of PI3K/AKT signaling [57]. Higher microtubule-affinity regulating kinase 2 (MARK2) expression was associated with the poor prognosis of OS and positively correlated with cisplatin chemoresistance mediated by the activation of the PI3K/AKT/NF-κB signaling pathway [58]. Also, microRNA-22-mediated cisplatin resistance is downstream from the PI3K/AKT/mTOR pathway in OS [59]. EGFR-facilitated tumor progression and gemcitabine resistance in OS were mediated through the PI3K/AKT pathway [60]. Moreover, proanthocyanidin B2 (PB2) inhibited the proliferation and induced the apoptosis of OS cells by suppressing the PI3K/AKT signaling pathway [61], whereas aclidinium bromide inhibited OS cell growth by regulating the PI3K/AKT signaling pathway [62]. As is evident, the PI3K/Akt signaling pathway significantly contributes to the development, progression, and therapeutic responsiveness of OS. The development of drugs targeting PI3K signaling has received much attention; indeed, enormous efforts have been dedicated and several are employed in evaluating clinical trials. The biggest hurdle in developing drugs targeting this pathway is the failure to generate a long-term outcome [63]. Cancer cells can compensate for the alternative pathway and acquire treatment resistance through feedback loops and crosstalk mechanisms. Another challenge is the lack of reliable predictive biomarkers that can identify patients who will most likely benefit from these types of therapies. The future development of promising inhibitors focusing on combined strategies, including the concomitant or sequential blockade of signaling pathways or the generation of less toxic drugs, will help develop novel therapeutic interventions for OS patients. 

### 3.2. JAK/STAT Signaling

There are four members of the JAK family, JAK1, JAK2, JAK3, and TYK2, and seven members of the STAT family, STAT1, STAT2, STAT3, STAT4, STAT5a, STAT5b, and STAT6 [64]. The JAK/STAT signaling pathway has recently gained much attention in relation to malignancies and autoimmune diseases. It regulates most immune regulatory processes, including those involved in tumor cell recognition and tumor-driven immune escape [65]. Antitumor immune responses are largely driven by the STAT1 and STAT2 induction of type I and II interferons (IFNs) and the downstream programs that IFNs potentiate. Conversely, STAT3 has been widely linked to cancer cell survival, immunosuppression, and sustained inflammation in the tumor microenvironment. Interleukin-6 (IL-6) is highly expressed in OS and is considered one of the JAK/STAT pathway activators [66]. Activation of the IL-6/JAK/STAT3 signaling pathway in cancer modulates the expression of several genes that drive the proliferation, metastasis, and survival of tumor cells while suppressing the antitumor immune response. The JAK/STAT pathway is also activated by other inducers in OS. Microarray and bioinformatics analysis shows a higher expression of serglycin, a potential biomarker of OS, which promotes OS proliferation, migration, and invasion by activating the JAK/STAT signaling pathway [67]. Similarly, FAT10 mediates OS development by activating the JAK/STAT signaling pathway [68]. STAT3 overexpression is negatively associated with OS’s five-year overall survival [69].

Several long non-coding RNAs altered in OS mediate tumor progression and metastasis through modulating the JAK/STAT pathway. A microarray analysis showed that *LINC01116* promoted OS progression by regulating the IL6R-JAK/STAT signaling pathway [70]. Overexpression of microRNA-101 inhibited OS tumor growth and metastasis by inactivating the PI3K/AKT and JAK/STAT signaling pathways mediated through the downregulation of ROCK1 [71]. Similarly, microRNA-126 inhibits proliferation, migration, invasion, and EMT in OS by targeting ZEB1 through the inactivation of the JNK and JAK1/STAT3 pathways [72].

Studies suggested that the blocking of the JAK/STAT pathway is a potential option for OS treatment [73]. Many JAK inhibitors have achieved efficacy in clinical settings, and more medications are currently being studied, as shown in Table 1. Suppressing STAT5 signaling affects OS growth and stemness. Curculigoside (Cur), a natural component of Curculigo orchioides Gaertn, controls OS growth by downregulating the JAK/STAT and NF-κB pathways, which is an underlying therapeutic option for OS treatment [74]. Similarly, curcumin and the curcumin analog L48H37 suppress human OS cell migration and invasion via inhibition of uPA, as mediated through the JAK/STAT signaling pathway [75,76]. Other targeted agents include the JAK2 inhibitors AG490 and telocinobufagin (TCB), which reduce OS cell proliferation, migration, and invasion by inhibiting the JAK2/STAT3 pathway in in vitro and in vivo set-ups [77]. 

### 3.3. Wnt/β-Catenin Signaling Pathway

Wnt/β-catenin is an evolutionarily conserved pathway regulating cell fate determination, cell migration, cell polarity, neural patterning, and organogenesis during embryonic development [78]. Mutations or alterations in the Wnt signaling pathway have been reported in several cancers and mediate tumorigenesis by modulating both tumors’ intrinsic and extrinsic properties in the tumor microenvironment mediated through crosstalk between transformed cells and infiltrating immune cells, such as leukocytes [79]. So far, three Wnt signaling pathways have been defined: the canonical Wnt pathway (Wnt/β-catenin), the non-canonical Wnt/PCP pathway (planar cell polarity), and the Wnt/Ca^2+^ pathway. 

The Wnt/β-catenin pathway is activated via the binding of extracellular Wnt ligands to transmembrane receptors by autocrine/paracrine methods. Once activated, the β-catenin is translocated to the nucleus, ultimately facilitating gene expression in cell proliferation, survival, differentiation, and migration [80]. Several reports showed the involvement of the Wnt/beta-catenin signaling pathway in OS tumor development and metastasis [81]. The SP1/Wnt/β-catenin signaling pathway plays a vital role in GABPB1-AS1-mediated OS tumorigenesis [82]. Melittin inhibits lung metastasis of human OS mediated by the Wnt/β-catenin signaling pathway [83]. Chemokine receptor-9 promotes EMT by activating the Wnt/β-catenin pathways to promote OS metastasis [84]. Expression of COL5A2 is elevated in OS patients, and its inhibition suppresses OS cell invasion and metastasis mediated through the TGF-β signaling and Wnt/β-catenin signaling pathways [85]. 

Several Wnt/β-catenin regulated non-coding RNAs are involved in osteosarcoma development and metastasis. Long non-coding RNA gastric carcinoma proliferation enhancing transcript 1 (lncGHET1) expression was significantly upregulated in OS cell lines, promoting OS development and progression via activating the Wnt/β catenin signaling pathway [86]. Similarly, lncLINC01128 regulates OS development by sponging microRNA-299-3p to mediate MMP2 expression and activate the Wnt/β-catenin signaling pathway [87]. Long non-coding RNA MINCR governs the growth and metastasis of human OS cells via the Wnt/β-catenin signaling pathway [88]. Overexpression of microRNA-135b, an oncogenic microRNA in OS, can promote OS invasion and metastasis in vitro and in vivo by activating the Wnt/β-catenin signaling pathway via directly targeting GSK-3β, APC, β-TrCP, and CK1α [89]. MicroRNA-340-5p suppresses osteosarcoma development by downregulating the Wnt/β-catenin signaling pathway via targeting the *STAT3* gene [90]. The circulating RNA Hsa_circ_0087302 affects the progression of osteosarcoma by modulating the Wnt/β-catenin signaling pathway [91]. 

The Wnt/β-catenin signaling pathway is widely activated in OS and highly related to the invasion and metastasis of OS. Therefore, an increasing number of studies focusing on Wnt/β-catenin signaling pathway inhibition showed a good response. We investigated the antitumor activity of tegavivint, a novel β-catenin/transducin β-like protein 1 (TBL1) inhibitor, against different OS models (in vivo/in vitro/ex vivo). It showed antiproliferative activity against OS cells in vitro and actively reduced micro- and macro-metastatic development ex vivo. We also observed that inherent chemoresistance was suppressed by tegavivint in vivo [92]. Tegavivint is a promising therapeutic agent for the advanced stages of OS, and currently, it is in clinical trials phase I/II. Echinatin inhibits the growth and metastasis of human osteosarcoma cells by regulating the Wnt/β-catenin and p38 signaling pathways [93]. 

### 3.4. NOTCH Signaling Pathway

NOTCH signaling is an evolutionarily conserved pathway associated with normal embryonic development, and it is dysregulated in several tumors, including OS [94]. The NOTCH signaling pathways consist of four receptors, including NOTCH-1, -2, -3, and -4, activated by a unique process that includes ligand binding and multistep proteolytic processing. The NOTCH ligands include Delta-like (DLL)-1, -3, -4, Jagged 1, and Jagged 2 in mammals. After binding NOTCH, the intracellular domain of the NOTCH ligand is ubiquitinated via the E3 ligase mind bomb-1, which initiates endocytosis of the NOTCH ligand/NECD complex into the ligand-expressing cell. The forces generated by these endocytosis-related events cause sequential proteolytic cleavage of NOTCH, which allows the NICD to be released into the cytosol and translocated to the nucleus. Higher expression of NOTCH3 and HES1, a downstream target of NOTCH signaling, is correlated with poor OS patient outcomes [95]. The NOTCH target genes, *HES1* and *HEY2*, were significantly upregulated in OS patient samples at the transcript level and associated with a poor prognosis [96]. JAG1 was found to be involved in the activation of various NOTCH receptors, and it is positively associated with the metastasis and recurrence of OS [97]. The NOTCH signaling pathway not only promotes OS tumor progression but is also involved in tumor metastasis. The expressions of NOTCH1, NOTCH2, HES1, and DLL1 were significantly upregulated in a highly invasive and metastatic LM7 cell compared to normal human osteoblasts and the SaOS-2 cell line, which have lower metastatic potential [98,99]. JAG1 expression was significantly higher in the highly metastatic F5M2 cell lines compared to the less metastatic F4 OS cells [97]. NOTCH3 could also mediate the invasion and metastasis of OS cells by upregulating the downstream target genes *HES1* and *MMP7* [95].

Cell migration-inducing protein (CEMIP) was overexpressed in OS tissues compared to non-tumor tissues, and its expression was positively associated with a poor prognosis. CEMIP promoted OS growth and metastasis by activating the NOTCH/JAG1/HES1 signaling pathway both in vivo and in vitro [100]. NOTCH-targeted therapy has shown potential for the treatment of OS in clinical research. CB-103 is a small molecule protein–protein interaction (PPI) inhibitor able to target the assembly of the NOTCH transcription complex and downregulate NOTCH target genes (*c-MYC*, *CCND1*, *HES1*) and the inhibition of NOTCH signaling. CB-103 has advanced into the first-in-human phase 1–2A study targeting pan-NOTCH signaling in advanced solid tumors and blood malignancies [101]. RO4929097, a NOTCH signaling inhibitor, was well tolerated by solid tumor patients, including sarcoma, and showed clinical antitumor activity, meaning further studies are warranted [102]. 

### 3.5. Receptor Tyrosine Kinase Pathway 

Protein kinases are enzymes that catalyze the transfer of a phosphate group from ATP to target proteins and play a crucial role in signal transduction and other cellular processes. Receptor tyrosine kinases (RTKs) are cell-surface growth-factor receptors with tyrosine-kinase activity. They regulate critical cellular processes, including cell proliferation, differentiation, cell survival, metabolism, and cell migration [103,104]. There are 58 known RTKs in humans, grouped into 20 subfamilies. All the RTKs have a similar molecular architecture, with a glycosylated N-terminal extracellular domain with a high number of disulfide bonds, which are involved in the ligand recognition, a cytoplasmic region that contains the protein tyrosine kinase (TK) domain, and a single transmembrane helix that plays a key role in the formation and stabilization of the dimer of the receptor chains. Several tyrosine kinase receptors have been associated with OS development and metastasis, including vascular endothelial growth factor receptor (VEGFR), platelet-derived growth factor receptor (PDGFR), insulin-like growth factor receptor (IGF), fibroblast growth factor receptor (FGFR), and AXL receptor tyrosine kinase (AXL). AXL RTK is highly expressed in most OS tissues and cell lines, and its higher mRNA expression correlates with poor clinical outcomes [52,105]. The knockdown of AXL inhibits the proliferation and induces the apoptosis of OS cells [106]. An increased expression of VEGF is associated with a lower overall OS patient survival [107]. However, using bevacizumab, a VEGF inhibitor, showed unfavorable outcomes [108], suggesting VEGF/VEGFR is limited and should be combined with the blockade of other pathways. FGFR amplification has been associated with OS resistance to chemotherapy [109] and the development of lung metastases [110]. Several RTKs inhibitors, including cabozantinib, bevacizumab, apatinib, sorafenib, and many more listed in Table 1, have shown promising effects and are at different stages of clinical trials. 

### 3.6. RANK/RANKL/OPG Pathway

The receptor activator of the nuclear factor κB ligand (RANKL) is a homotrimeric transmembrane protein member of the tumor necrosis factor (TNF) cytokine family expressed by osteoblast cells and tumor cells. In contrast, RANK and osteoprotegerin (OPG) are transmembrane receptor members of the TNF receptor (TNFR) family expressed on the surface of osteoclasts. The RANK/RANKL signaling pathway is crucial for bone homeostasis, osteoclast survival, differentiation, and function. OPG counterbalances RANKL and prevents RANK/RANKL interaction [111]. Several studies have shown the dysregulation of this pathway in OS and its association with metastasis and chemoresistance [112]. Numerous human OS cell lines express higher levels of RANKL, and the ratio of RANKL/OPG was shifted in favor of RANKL in blood samples derived from OS patients [113,114]. RANK was negatively associated with patient survival and chemotherapy response in 91 human samples [115]. OPG administration in OS mouse models indirectly affected tumor progression, diminished tumor growth, and increased survival [116]. Punzo et al. tested denosumab, a human monoclonal antibody with high binding affinity and specificity to RANKL, alone and in combination with doxorubicin but found a discouraging response as it worsened the effect of standard chemotherapy [117]. As reported previously, osteoclast cell abundance in the OS tumor microenvironment prevents metastasis as well as improves the chemotherapeutic response. Additional preclinical studies are needed before blocking the RANK/RANKL pathways in OS. 

### 3.7. Hedgehog/Gli Signaling Pathway

Hedgehog (Hh)/Gli signaling is a conserved signal transduction pathway with a key regulatory function in physiological processes, including embryonic development, tissue differentiation, and cell growth. Dysregulation of the Hedgehog/Gli pathway is considered a critical factor in the development and progression of multiple cancers. Several Hedgehog/GLI pathway-associated genes are candidate targets for tumor therapy [118]. The pathway is unique in that it is comprised of both tumor suppressor genes and oncogenes. The signaling pathway is associated with three ligands, Sonic hedgehog (SHH), Indian hedgehog (IHH), and Desert hedgehog (DHH), and additional components of the pathway include 12-transmembrane patched proteins (PTCH1 and PTCH2), 5-zinc finger transcription factors GLI1, GLI2, GLI3 (glioma-associated oncogene homologs), and the 7-transmembrane protein smoothened (SMO). The Hh/Gli pathway can promote OS metastasis by interacting with other signaling pathways, such as the PI3K/AKT and Wnt pathway.

Lo et al. evaluated the Hh pathway in 42 human OS samples and found higher expression levels of genes encoding the IHH, PTCH1, and GLI in the tumors [119]. Hirotsu et al. demonstrated that SHH, DHH, PTCH1, GLI1, GLI2, and SMO were overexpressed human OS cell lines [120]. They also reported that SMO and GLI activation are vital for OS progression and that their inhibition suppresses OS cell proliferation both in vivo and in vitro [121]. Hedgehog inhibition also prevented migration and metastasis in mouse models [122]. Several Hh-pathways inhibitors, including the SMO inhibitors (cyclopamine, IPI-926 (saridegib), GDC-0449 (vismodegib), and LDE225 (erismodegib), and Gli inhibitors (arsenic trioxide (ATO), Gli antagonists (GANTs)) have been assessed in preclinical models for OS [123]. Furthermore, taladegib is presently in a phase II clinical trial for advanced solid tumors with PTCH1 mutations (NCT05199584). However, more research is needed to discover the broad biological effects of Hedgehog pathway inhibition on OS tumor development and metastasis. 

### 3.8. Transforming Growth Factor-β Signaling Pathway

Transforming growth factor-β (TGF-β) represents an evolutionarily conserved family of secreted polypeptide factors that mediate a diverse range of embryonic and adult signaling functions. In contrast with the dual effects of TGF-βs on tumor progression, TGF-βs seem to have a pro-tumoral effect on sarcomas, specifically in OS. TGF-βs expression is increased in OS patient sera compared to the healthy donors [124] and associated with lung metastases [125,126]. The higher expression of TGF-βs in OS shows a poor response to chemotherapy [127]. COL5A2, which is highly expressed in metastatic OS patients and associated with poor outcomes, is mediated through the TGF-β and Wnt/β-catenin signaling pathways [85]. The IHC staining and qPCR analysis of OS patient samples showing higher expression of the six-transmembrane epithelial antigen of prostate 1 (STEAP1) correlated with poor outcomes is also mediated through the Wnt/β-catenin and TGF-β/Smad2/3 pathways [128]. OS patients with high serum levels of growth and differentiation factors 15 (GDF15) exhibited significantly decreased overall survival and pulmonary metastasis-free survival (PMFS). The knockdown of GDF15 attenuated the migration and invasion of OS cells mediated through the TGF β signaling pathway [129]. TGF-β has been found to protect OS cells from chemotherapeutic cytotoxicity in a succinate dehydrogenase (SDH)/HIF1α-dependent manner [130]. Gamabufotalin, a natural derivative of the Chinese medicine Chansu, suppressed OS stem cells through the TGF-β/periostin/PI3K/AKT pathway [131]. MicroRNA-181c suppresses OS cell progression by targeting SMAD7 and regulating the TGFβ signaling pathway [132]. microRNA-522 was highly expressed in OS cells and presented carcinogenic function by contributing to cell proliferation, migration, and EMT progression mediated through the TGF-β/Smad pathway [133]. Several TGF-β inhibitors are being used in the preclinical set-up, i.e., RepSox, a TGF-β inhibitor, suppresses OS proliferation and EMT and promotes apoptosis by inhibiting the JNK/Smad3 signaling pathway. Oridonin, a bioactive diterpenoid, inhibits EMT and TGF-β1-induced EMT by inhibiting the OS’s TGF-β1/Smad2/3 signaling pathway [134]. The TGF-βR1 inhibitor vactosertib significantly inhibited OS proliferation in vitro and in vivo. It repressed c-Myc expression, inhibited immune suppressor cells (M2-like TAM, MDSC), and enhanced immune effectors (IFNγ + CD8 + cells and NK cells) in the OS tumor microenvironment [135].

## 4. Conclusions and Future Directions

OS is the most common and highly metastatic primary bone tumor in children. The long tubular bones are the most commonly affected bones, along with the spine, pelvis, and sacrum areas. This tumor is highly metastatic and invasive, and the primary site of metastasis is the lung, with a high mortality rate. Contrasting with many other sarcomas, OS lacks an established translocation or genetic mutation, and the causative factor in most cases remains unclear. Clinical outcomes for this disease have not progressed in over 30 years due to its complex and heterogeneous nature. The progressive accumulation of *TP53* and the *RB* mutation (tumor suppressor genes) are currently attributed to the disease development. Several tumor-promoting genes, including MYC and MDM2, are thought to be involved in the disease progression. We and others are trying to understand the role of MYC in OS disease progression and chemoresistance. We recently developed a Myc knock-in mouse model and are trying to understand its role in disease progression. 

The current standard care for OS management is surgical excision and systemic multiagent therapy. However, it is insufficient and there is a pressing need to identify novel therapeutic approaches for this highly aggressive and metastatic disease. Signaling pathways are highly dysregulated and contribute to OS development, metastasis, and chemoresistance. Enormous efforts have been dedicated to identifying the development of new drugs targeting signaling pathways. Several drugs targeting these pathways are in clinical trials and show promising effects, as shown in Table 1. The challenge associated with these targeted therapies is that cancer cells can compensate for the alternative routes and acquire treatment resistance through feedback loops and crosstalk mechanisms. Developing drugs focusing on combined strategies, including the concomitant or sequential blockade of signaling pathways or the generation of less toxic drugs, will help develop novel therapeutic interventions for OS patients. Immunotherapy has gained much attention in relation to cancer treatment but is still ineffective in OS disease management. The poor infiltration of the immune cells to the tumor microenvironment (TME) of OS, low activity from the available T cells, a lack of immune-stimulating neoantigens, and activation of multiple immune-suppressing pathways all combine to dampen responses to immunotherapy. Several approaches are currently being tested, including enhancing NK cells’ mediated natural immunity and using IL-2 to activate the effector T cells, as examples to enhance the efficacy of immunotherapy. Several other approaches, including non-specific adoptive T cell immunotherapy and CART therapy, are also being tried in preclinical and clinical practices. Non-specific immunotherapies stimulate or boost the immune system but do not target cancer cells directly through the use of certain cytokines, including IL2, IL-7, IL-12, IL-21, and IFN-α/β/γ, and the checkpoint inhibitors PD-1, PD-L1, and CTLA-4, and immuno-stimulatory agents such as CpG oligonucleotides, and agonistic CD40 are an example of nonspecific immune therapy and have sometimes led to a better immune response against cancer cells. In adoptive T-cell (ATC) therapy, autologous or allogenic T-cells are infused into patients with cancer, which has shown considerable promise in recent years. Recently, CART therapy has generated substantial excitement among researchers and oncologists. In CART therapy, T-cells are collected from the patient and re-engineered in the laboratory to produce proteins on their surface called chimeric antigen receptors, or CARs. The CARs can identify and bind to the antigens on the surface of cancer cells. Chemotherapy in combination with immune modulators and checkpoint inhibitors represents an attractive area of research and shows promising effects in OS management. Table 1 shows past and running clinical trials with immunotherapy or combination therapies. Much work is still needed to develop a novel effective immunotherapy against the OS tumor. We and others strongly believe there is much potential in immunotherapy that can revolutionize the treatment of OS in the future.

## Figures and Tables

**Figure 1 ijms-24-11367-f001:**
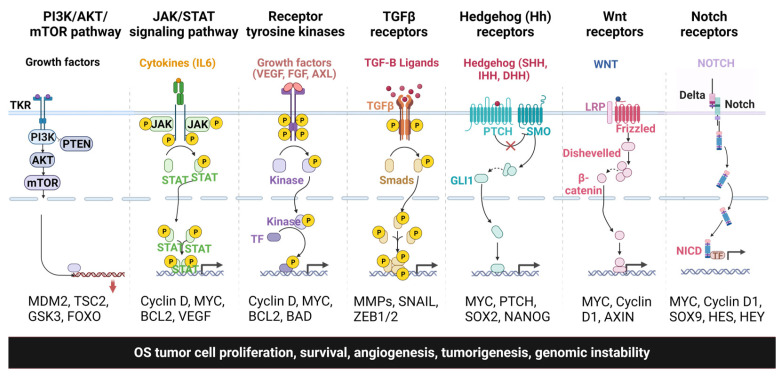
Different signaling pathways involved in OS tumor progression, metastasis, and chemoresistance.

**Table 1 ijms-24-11367-t001:** List of signaling pathway inhibitors and immunotherapeutic agents in clinical trials.

Name of Compound	Target Pathway	Mechanism of Action	Phase	NCT Number
Cabozantinib + chemotherapy	RTK	VEFGR2/MET inhibitor	II/III	NCT05691478
Bevacizumab (Avastin^®^) + cisplatin, doxorubicin, and high-dose methotrexate	RTK	VEGF	II	NCT00667342
Apatinib + gemcitabine-docetaxel chemotherapy	RTK	VEFGR inhibitor	II	NCT03742193
Ifosfamide and etoposide with or without lenvatinib	RTK	VEGFR 1/2/3 FGFR 1/2/3/4, PDGFR/alpha, c-Kit, and the RET proto-oncogene	II	NCT04154189
Sorafenib	RTK	RAF1, BRAF, VEGFR 1, 2, 3, PDGFR, KIT, FLT3, FGFR1, and RET	II	NCT00880542
Surufatinib	RTK	VEGFR1, VEGFR2, VEGFR3, FGFR-1, and CSF-1R	II	NCT05106777
Erlotinib in combination with temozolomide	RTK	EGRF	II	NCT02689336
Temozolomide and irinotecan hydrochloride with or without bevacizumab	RTK	VEGF	II	NCT01217437
Sunitinib and/or nivolumab plus chemotherapy	RTK	VEGFRs, PDGFR and the MAPK pathway	I/II	NCT03277924
Lenvatinib as a single agent and in combination with chemotherapy (ifosfamide and etoposide)	RTK	VEGFR-1 (FLT1), VEGFR-2 (KDR), VEGFR-3 (FLT4), FGFR-1, FGFR-2, FGFR-3, FGFR-4, PDGFRa, RET, and c-KIT.	I/II	NCT02432274
Bevacizumab	RTK	VEGF inhibitor	I	NCT00620295
Cabozantinib with topotecan-cyclophosphamide	RTK	VEGFR-1, -2 and -3, KIT, TRKB, FLT-3, AXL, RET, MET, and TIE-2	I	NCT04661852
Erlotinib	RTK	EGRF	I	NCT00012181
Ridaforolimus	PI3K/AKT/mTOR	mTOR inhibitor	III	NCT00538239
Everolimus (RAD001)	PI3K/AKT	mTOR	II	NCT01830153
Chemotherapy with or without trastuzumab	PI3K	HER2	II	NCT00023998
Robatumumab	PI3K/AKT	IGF-1R	II	NCT00617890
Cyclophosphamide and sirolimus (OCR)	PI3K/AKT/mTOR	mTOR	II	NCT00743509
Sirolimus in combination with metronomic	PI3K/AKT/mTOR	mTOR	II	NCT02574728
R1507	PI3K/AKT	IGF1	II	NCT00642941
Ganitumab (AMG 479)	PI3K/AKT	IGF1R	II	NCT00563680
TQB3525	PI3K/AKT/PTEN	PI3Ka and PI3Kd inhibitor	I/II	NCT04690725
ABT-751	PI3K	Inhibiting the binding of VEGF to its cell surface receptors	I	NCT00036959
Cixutumumab and doxorubicin hydrochloride	PI3K/AKT	IGF1R	I	NCT00720174
Palbociclib combined with chemotherapy	CDK	CDK4/6	II	NCT03709680
Alvocidib	CDKs	Cyclin D1 and D3	I	NCT00012181
CB-103	NOTCH	Pan NOTCH pathway inhibitor	I/II	NCT03422679
RO4929097	NOTCH	Notch signaling pathway inhibitor	I/II	NCT01154452
RO4929097	NOTCH	Vessel growth	I	NCT01236586
Cicatricial alopecia	JAK/STAT	JAK1/TYK2 inhibitor	II	NCT05076006
ENV-101 (taladegib)	Hedgehog	Hedgehog pathway inhibitor	II	NCT05199584
Tegavivint	Wnt/β-catenin signaling	Interaction between β-catenin and transducin β-like protein 1 (TBL1)	I/II	NCT04851119
Vactosertib	TGF-β	(TGF-β) type 1 receptor inhibitor	I/II	NCT05588648
Denosumab	RANK/RANKL	Inhibit RANK binding	II	NCT02470091
Apatinib mesylate plus anti-PD1 therapy	RTK	VEGFR tyrosine kinase inhibitor	II	NCT03359018
Camrelizumab (PD1) with neoadjuvant chemotherapy	Checkpoint inhibition	PD1	II	NCT04294511
CART plus chemotherapy	Immunotherapy	CAR-T cells and sarcoma vaccines	I/II	NCT04433221
Haploidentical transplant and donor natural killer cells	Immunotherapy	NK cell activation	II	NCT02100891
Avelumab	Immunotherapy	PDL1	II	NCT03006848
Atezolizumab and cabozantinib	Immunotherapy	Immune checkpoint inhibitor	II	NCT05019703
GD2-targeted modified T-cells (GD2CART)	Immunotherapy	CART	I	NCT04539366
Famitinib plus camrelizumab	RTK and checkpoint inhibitor	VEGFR-2, -3 and FGFR-1, -2, -3, -4 and PD1/PDL1	II	NCT04044378
B7H3 CAR T-cell	Immunotherapy	CART, B7H3-specific receptor	I	NCT04483778
EGFR806 CAR T-cell	Immunotherapy	EGFR	I	NCT03618381
Pepinemab	PI3K/AKT	SEMA4D inhibitor	I/II	NCT03320330
Activated T-cells armed with GD2 bispecific antibody	Immunotherapy	CART	I/II	NCT02173093
Cytokine-induced killer (CIK)	Adoptive immunotherapy	Cytokine-induced killer	I	NCT03782363
Recombinant oncolytic herpes simplex virus type I (R130)	Oncolytic virus injection	Oncolytic herpes simplex virus type	I	NCT05851456
Sarcoma-specific CAR-T-cells	Immunotherapy	CART	I/II	NCT03356782
Oleclumab plus durvalumab	Immunotherapy	Anti-CD73 monoclonal antibody and PD-1	II	NCT04668300
Nivolumab plus ipilimumab	Immunotherapy	PD1 and CTLA-4	II	NCT02500797
GD2-targeted modified T-cells (GD2CART)	Immunotherapy	CART	I	NCT02107963
Nivolumab plus ipilimumab	Immunotherapy	PD1 and CTLA-4	I/II	NCT02304458
Vigil	Engineered autologous tumor cell immunotherapy	GM-CSF, TGFβ-1 and TGFβ-2		NCT03842865
B7H3 CAR T-cell	Immunotherapy	CSRT, B7H3-specific receptor	I	NCT04897321
Haploidentical donor NK cells and Hu14.18-IL2	Immunotherapy	NK cell activation	I	NCT03209869
Pembrolizumab	Immunotherapy	PD1	II	NCT02301039
Nivolumab plus ipilimumab	Immunotherapy	PD1 and CTLA-4	II	NCT02982486
Multi-component immune-based therapy (MKC1106-PP)	Immunotherapy	T-cell	I	NCT00423254
Durvalumab plus tremelimumab	Immunotherapy	PD1 and CTLA-4	II	NCT02815995
Nivolumab plus ipilimumab	Immunotherapy	PD1 and CTLA-4	II	NCT05302921
C7R-GD2.CART cells	Immunotherapy	CART	I	NCT03635632
Tislelizumab	Immunotherapy	PD1	II	NCT05241132
CAB-AXL-ADC plus PD-1 inhibitor	Immunotherapy	PD1	I/II	NCT03425279

## Data Availability

Not applicable.

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
