# Peer review of "Deciphering the Signaling Mechanisms of Osteosarcoma Tumorigenesis"

_ijms, 2023, doi:10.3390/ijms241411367_

Round 1
Reviewer 1 Report
The idea to review signaling mechanisms of osteosarcoma (OS) tumorigenesis is very important and quite interesting as it might help in the search of a therapeutic target for the disease which has complex and heterogenous nature.
Presence of a graphical abstract in the Review helps to catch the main idea at a glance which is a benefit.
The provided information is analyzed and logically structured, the presence of the Figure which illustrates different signaling pathways in osteosarcoma tumor progression, metastasis, and chemoresistance makes the information in the text more clear.
Description of the signaling pathways is given and their involvement into the OS progression is characterized by Authors.
List of signaling pathway inhibitors and immunotherapeutic agents in clinical trials is provided in the Review which is very informative and helps to understand the current situation in the field.
Based on the analysis of the existing data in the area Authors offered further directions of the research.
The following comments do not diminish the value of the Review:
Line 26 Should the Graphical abstract be counted as a Figure?
Line 110 It would be better to remove the dot at the end of the tile: ‘3. Major signaling pathways in OS tumor development and metastasis.’.
Line 178 In the text there is a mention of Figure 1 (line 115), the information in the text or the number of the current Figure 2 should be corrected.
Line 178 The font of the text in Figure 2 should be changed a bit probably, so the information would become visible.
Line 420 It would be better to decipher the following abbreviature: ‘TME’.
Author Response
Reviewer 1
Line 26 Should the Graphical abstract be counted as a Figure?
We appreciate the reviewer's suggestions and counted the graphical abstract as Figure 1 in the manuscript.
Line 110 It would be better to remove the dot at the end of the tile: ‘3. Major signaling pathways in OS tumor development and metastasis.’.
We appreciate the reviewer's suggestions and changed it accordingly.
Line 178 In the text there is a mention of Figure 1 (line 115), the information in the text or the number of the current Figure 2 should be corrected.
We appreciate the reviewer's suggestions and changed it accordingly.
Line 178 The font of the text in Figure 2 should be changed a bit probably, so the information would become visible.
We appreciate the reviewer's suggestions and modified the figure text.
Line 420 It would be better to decipher the following abbreviature: ‘TME’.
We appreciate the reviewer's suggestions and put TME abbreviation in the text.
Reviewer 2 Report
1. The work of Nirala and colleagues is an interesting and highly detailed review manuscript on the dysregulated mechanism relying on the onset and development of osteosarcoma. So, the ms potentially fits adequately with the topic of International Journal of Molecular Sciences MDPI. With exceptions, the review comprehensively cover in detail the implications of major mutations, chromosomal amplifications/loss and gene expression dysregulations in the tumorigenic process of osteosarcoma. Numerous aspects of this topic are extensively covered. Figures are fine but their quality should be improved. The table summarizing the clinical trials is well done and informative. Please see below my comments/minor observations for improving the work:
2. A couple of words on the main epigenetic dysregulation might be helpful for the reader. Authors can check PMID: 26349415 and https://doi.org/10.3390/cells12121595
3. It is unclear the meaning of the “S.N” column of table 1
4. Section 3.1 PI3K/Akt dysregulation has also been reported in other tumors, such as skin cancers and brain tumors (https://www.nature.com/articles/s41388-021-02090-z and https://www.ncbi.nlm.nih.gov/pmc/articles/PMC5078108/ . For completeness, this information and supporting reference should be included.
5. Several Hedgehog/GLI pathway associated genes are currently candidate targets for tumor therapy (PMID: 35163655). What about OS?
6. Please improve figures quality
Author Response
Reviewer 2
- A couple of words on the main epigenetic dysregulation might be helpful for the reader. Authors can check PMID: 26349415 and https://doi.org/10.3390/cells12121595
We appreciate the reviewer's suggestions and incorporated a paragraph on epigenetic dysregulation in OS progression in section 2.3 in the text.
- It is unclear the meaning of the “S.N” column of table 1
We appreciate the reviewer's suggestions and after further consideration we have removed the S.N. column from the Table. It did not provide any value to the Table.
- Section 3.1 PI3K/Akt dysregulation has also been reported in other tumors, such as skin cancers and brain tumors (https://www.nature.com/articles/s41388-021-02090-z and https://www.ncbi.nlm.nih.gov/pmc/articles/PMC5078108/ . For completeness, this information and supporting reference should be included.
We appreciate the reviewer's suggestions and incorporated this information into the manuscript.
- Several Hedgehog/GLI pathway associated genes are currently candidate targets for tumor therapy (PMID: 35163655). What about OS?
We appreciate the reviewer's suggestions and incorporated information regarding Hedgehog inhibitors into the manuscript.
- Please improve figures quality
We appreciate the reviewer's suggestions and we improved the figure text and image resolution.